# The alteration of LBX1 expression is associated with changes in parameters related to energy metabolism in mice

**Takahiro Nakagawa**[1], **Keisuke Horiuchi**[1]\*, **Kazuki Kagami**[2], **Shinya Kondo**[1], **Masashi Isaji**[1], **Yusuke Matsuhashi**[1], **Kazuya Kitamura**[1], **Takeshi Adachi**[2], **Kazuhiro Chiba**[1]

1 Department of Orthopedic Surgery, National Defense Medical College, Tokorozawa, Saitama, Japan,
2 Division of Cardiovascular Medicine, Department of Internal Medicine I, National Defense Medical College, Tokorozawa, Saitama, Japan

\* khoriuchi@ndmc.ac.jp

**Data Availability Statement:** All relevant data are within the manuscript and its Supporting Information files.

## Abstract

The LBX1 gene is located near a single nucleotide polymorphism that is highly associated with susceptibility to adolescent idiopathic scoliosis and is considered one of the strongest candidate genes involved in the pathogenesis of this condition. We have previously found that loss of LBX1 from skeletal muscle results not only in spinal deformity but also in lean body mass, suggesting a potential role for LBX1 in energy metabolism. The purpose of the present study was to test this hypothesis by analyzing the phenotype of mice lacking LBX1 in skeletal muscle with a focus on energy metabolism. We found that loss of LBX1 rendered mice more resistant to high-fat diet-induced obesity, despite comparable food intake between mutant and control mice. Notably, the mutant mice exhibited improved glucose tolerance, increased maximal aerobic capacity, and higher core body temperature compared to control mice. In addition, we found that overexpression of LBX1 decreased glucose uptake in cultured cells. Taken together, our data show that LBX1 functions as a negative regulator of energy metabolism and that loss of LBX1 from skeletal muscle increases systemic energy expenditure resulting in lean body mass. The present study thus suggests a potential association between LBX1 dysfunction and lean body mass in patients with adolescent idiopathic scoliosis.

## Introduction

Adolescent idiopathic scoliosis (AIS) is one of the most common spinal conditions that affects approximately 2–3% of school-aged children [1, 2]. The majority of patients do not require treatment; however, surgery is often performed to correct the deformity and prevent further progression of the condition in patients with severe curvature. Otherwise, there is currently no effective treatment for this condition. Although it is known that AIS is associated with both environmental and genetic factors, the etiology of AIS remains elusive. Given that surgical

**Funding:** JSPS KAKENHI - stands for "Grants-in-Aid for Scientific Research from Japan Society for the Promotion of Science" (22K09320). The funders had no role in study design, data collection and analysis, decision to publish, or preparation of the manuscript.

**Competing interests:** The authors have declared that no competing interests exist.

treatment is highly invasive and can lead to serious complications, it is imperative to understand the etiology of AIS and establish a preventive treatment for this condition.

Previous studies have identified several single nucleotide polymorphisms associated with susceptibility to AIS. Among these, rs11190870, a locus on chromosome 10q24.31, is the most significant and has been highly replicated in various studies [3, 4]. Although rs11190870 is not within the coding sequence of a gene, it is relatively close to the LBX1 gene, which encodes the ladybird homeobox 1 transcription factor [4]. In mice and humans, LBX1 is highly expressed in the spinal cord and skeletal muscle, and is thought to be one of the strongest candidate genes involved in the pathogenesis of AIS [4, 5]. Several studies have shown that LBX1 is involved in muscle development and neuronal determination; however, its role in muscle homeostasis and its potential involvement in the etiology of AIS are not fully understood [3]. To better understand the function of LBX1 and to explore the potential association of LBX1 with AIS, we developed mutant mice in which the LBX1 gene is conditionally ablated in skeletal muscle (hereafter referred to as $Lbx1^{\Delta Mus}$ mice). Using this mouse model, we previously reported that loss of LBX1 in skeletal muscle from embryogenesis leads to hypoplastic forelimbs and progressive kyphosis in mice [5]. Interestingly, in addition to these defects, we also found that $Lbx1^{\Delta Mus}$ mice are consistently leaner than their wildtype counterparts, reminiscent of AIS patients who are generally leaner than their age-matched control population [6–10]. Since the prevalence of eating disorders is lower in AIS patients than in control subjects, several studies have suggested that metabolic abnormalities, and the resulting lean physique, are potentially associated with the pathogenesis of AIS. Nevertheless, this coincidence of lean physique between $Lbx1^{\Delta Mus}$ mice and AIS patients led us to hypothesize that LBX1 may be involved in energy metabolism and that dysregulation of LBX1 expression could lead to leaner body mass.

In this study, we aimed to test this hypothesis by investigating the potential function of LBX1 in regulating energy metabolism. We found that $Lbx1^{\Delta Mus}$ mice are resistant to high-fat diet-induced obesity and have higher maximal aerobic capacity and core body temperature than control mice. Of note, despite the leaner physique of $Lbx1^{\Delta Mus}$ mice, food intake was comparable between $Lbx1^{\Delta Mus}$ and control mice, and total activity was lower in $Lbx1^{\Delta Mus}$ mice than in control mice. Furthermore, we also found that overexpression of LBX1 in cultured cells decreased glucose uptake in these cells. These findings reveal a previously unknown role of LBX1 as a negative regulator of energy metabolism and may suggest that dysregulation of LBX1 is associated with leanness in AIS patients.

## Materials and methods

### Mice

The generation of $Lbx1^{flox/flox}$ mice was previously described [5, 11]. The mice were crossed with $ACTA1^{cre/+}$ transgenic mice to conditionally remove the floxed-$Lbx1$ allele from skeletal muscle ($Lbx1^{\Delta Mus}$ mice). $Lbx1^{flox/flox}$ mice were used as control (Ctrl) animals. Mice were housed under specific pathogen-free conditions, maintained to a 12 h light/ 12 h dark cycle, and provided with standard chow and water. In some experiments, mice were fed high-fat chow (40% fat; Oriental Yeast, Tokyo, Japan) when they reached 5 weeks of age. Mice were monitored for behavioral and health status at least once a week. To reduce distress, all invasive procedures, except for toe clipping at 2 weeks of age, were performed under isoflurane inhalation anesthesia. Euthanasia was performed by cervical dislocation after isoflurane inhalation. All animal experiments were approved by the Animal Care Committee of the National Defense Medical College (approval number: 18078).

## Histology

Perigonadal adipose tissues from 30-week-old Ctrl and $Lbx1^{\Delta Mus}$ mice were fixed in 10% formalin neutral buffer solution and embedded in paraffin. Paraffin sections were stained with hematoxylin and eosin. The cross-sectional area of adipocytes, which reflects the degree of intracellular fat accumulation, was measured using Image J software (National Institutes of Health, Bethesda, MA).

## Intraperitoneal glucose tolerance test

11-week-old Ctrl and $Lbx1^{\Delta Mus}$ mice were injected intraperitoneally with glucose solution (1.5 g/kg) after overnight starvation (16 hours). Blood glucose concentrations were measured using the On Call Express Mobile Diabetes Testing Kit (ACON, San Diego, CA) at 0, 15, 30, 60, and 120 min after glucose injection.

## Assessment of maximum oxygen uptake ($VO_2$ max), basal $O_2$ consumption, respiratory quotient, and spontaneous activity

11-week-old Ctrl and $Lbx1^{\Delta Mus}$ mice were subjected to an incremental exercise test to assess $VO_2$ max on a 5-degree incline treadmill using the Oxymax System (Columbus Instruments, Columbus, OH). To familiarize the mice with the environment, the mice were placed in the apparatus for 30 min prior to the experiments. Mice were run on a treadmill starting at a speed of 10 m/min and increasing by 1 m/min every 1 min to 25 m/min. The experiments were stopped when the mice were in contact with the electrical grid for 5 s due to fatigue. Basal $O_2$ consumption and respiratory quotient were evaluated every 10 min using the Oxymax System (Columbus Instruments) and analyzed using Oxymax for Windows Software (Columbus Instruments). Spontaneous locomotor activity of the mice was evaluated every 10 min using an ACTIMO-100M monitoring system (Shinfactory, Fukuoka, Japan) and analyzed using ACTIMO-DATA software (Shinfactory). Mice were individually housed in each chamber and monitored for 42 h after a 6-h acclimation period. Diurnal and nocturnal periods were defined as 7:00 am to 7:00 pm and 7:00 pm to 7:00 am, respectively. Food intake was simultaneously assessed for 24 hours. Core body temperature was measured through the rectum.

## Gene expression analysis

Tibialis anterior muscles were collected from 11-week-old Ctrl and $Lbx1^{\Delta Mus}$ mice. Total RNA was extracted using Isogen II (Nippon Gene, Tokyo, Japan). Transcript expression was analyzed using an Affymetrix Mice Clariom S Array (Thermo Fisher Scientific, Waltham, MA). Differentially expressed genes were identified using Transcriptome Analysis Console software (Thermo Fisher Scientific). The expression of differentially expressed genes with a potential role in energy metabolism was verified by quantitative PCR. The nucleotide sequences of the PCR oligos used in the present study are shown in Table 1.

## Epitope-tagged mouse LBX1-expression vector

The full-length mouse $Lbx1$ transcript was cloned by PCR using cDNA prepared from skeletal muscle and inserted into the pcDNA 4/$myc$-His expression vector (Thermo Fisher Scientific) through the EcoRI and XbaI cloning sites. The epitope-tagged LBX1 (henceforth referred to as LBX1$^{myc}$) expression vector was transfected into the mouse fibroblast-like cell line NIH/3T3 (Riken BRC, Ibaragi, Japan) using Lipofectamine 3000 (Thermo Fisher Scientific). Expression of LBX1$^{myc}$ was confirmed by Western blot. Anti-myc tag antibody (1:1000) and HRP-conjugated anti-beta actin antibody (1:10000) were purchased from Merck (Rahway, NJ) and GE

**Table 1. Nucleotide sequences of oligos used in the present study.**

| Gene | Forward | Reverse |
| --- | --- | --- |
| Lbx1 | GCGATGGGATGACCATCTTT | GCGTTTCTCCAACTCCAACTCGTAGAT |
| Gapdh | AACAGCAACTCCCACTCTTC | CCTGTTGCTGTAGCCGTATT |
| Grb14 | CTGTGCTAGGATGGCTGTTTAG | AGCCTTGTGATCTCTCTCTCTC |
| Igfn1 | GGACATCTCGGCTTCTTGATAG | CTGGTCAGCAACTCTACATCTT |
| Mss51 | GACTTCTGGGAGGAGCAAATAG | GGGTAACCAAGCTTCCATCA |
| Myh7 | AGATGGCTGGTTTGGATGAG | TTGGCCTTGGTCAGAGTATTG |
| Bdh1 | CACTGTTCTAGCTCCTGTCTTC | CATCCCGCTGTCAGGTAAAT |
| Igfbp5 | CCTTGGTGTCTCCCATCTAATC | CTGTGACTTGAACTGACCTCAT |
| Irs2 | CTGCTGCTCACTTTCCTATCA | CCTGCCTCTTGGTTCCTTATC |
| Nr4a2 | GCACGTCAAAGAACTGGAAAG | GAAGGGAGATAAGGTGAACTGG |
| Nr4a3 | GGCCTCTCTTCTGTTCTTTCTT | CAACTGGCTGGACTACTTATGG |

Healthcare Biosciences (Chicago, Il), respectively. Glucose consumption of cells transfected with empty vector or LBX1$^{myc}$ expression vector was evaluated using the Glucose Assay Kit-WST (Dojindo, Kumamoto, Japan) following the manufacturer's instructions.

In brief, cells ($1.35 \times 10^5$ cells/well) were cultured in 2 ml of maintenance medium (DMEM supplemented with 10%fetal bovine serum and penicillin/streptomycin) in 12-well plates at 37˚C. After 16 h of culture, the media were collected and centrifuged at 1,500 rpm for 5 min. The supernatants were incubated with the reaction buffers provided by the manufacturer for 30 min at 37˚C, and the absorbance of the supernatant was measured at 450 nm. Glucose consumption was determined by subtracting the pre-culture glucose concentration from the post-culture glucose concentration. Total RNA was extracted from cultured cells using the RNeasy Plus Mini Kit (Qiagen, Venlo, Netherlands).

## Statistics

Student's *t*-test, paired t-test, Mann-Whitney U test, and two-way ANOVA with Sidak correction were used for statistical analysis. $P < 0.05$ was considered statistically significant. Data are presented as the mean ± standard error of the mean.

## Results

### Conditional ablation of *Lbx1* in skeletal muscle confers resistance to obesity in mice

In the previous study, we documented that *Lbx1$^{\Delta Mus}$* mice had a lower body weight than Ctrl mice up to 8 weeks of age. To further validate this observation, we investigated whether this result would hold in older mice. We housed the mice under normal conditions with regular chow and followed them until they reached 50 weeks of age. As shown in Fig 1A and 1B, we found that *Lbx1$^{\Delta Mus}$* mice consistently had a lower body weight on average than Ctrl animals. The difference in body weight was greater in male than in female mice. We next fed a high-fat diet to determine whether *Lbx1$^{\Delta Mus}$* mice would be resistant to obesity. As expected, the body weight of both *Lbx1$^{\Delta Mus}$* and Ctrl mice increased significantly with age compared to those fed regular chow; however, the increase was significantly reduced in *Lbx1$^{\Delta Mus}$* mice (Fig 1C). Notably, we found that the difference in weight gain between *Lbx1$^{\Delta Mus}$* and Ctrl mice was significantly greater in female (Ctrl, 58.0 g; *Lbx1$^{\Delta Mus}$*, 39.8 g; 18.2 g difference) than in male mice (Ctrl, 56.4 g; *Lbx1$^{\Delta Mus}$*, 45.2 g; 11.2 g difference) at 50 weeks of age.

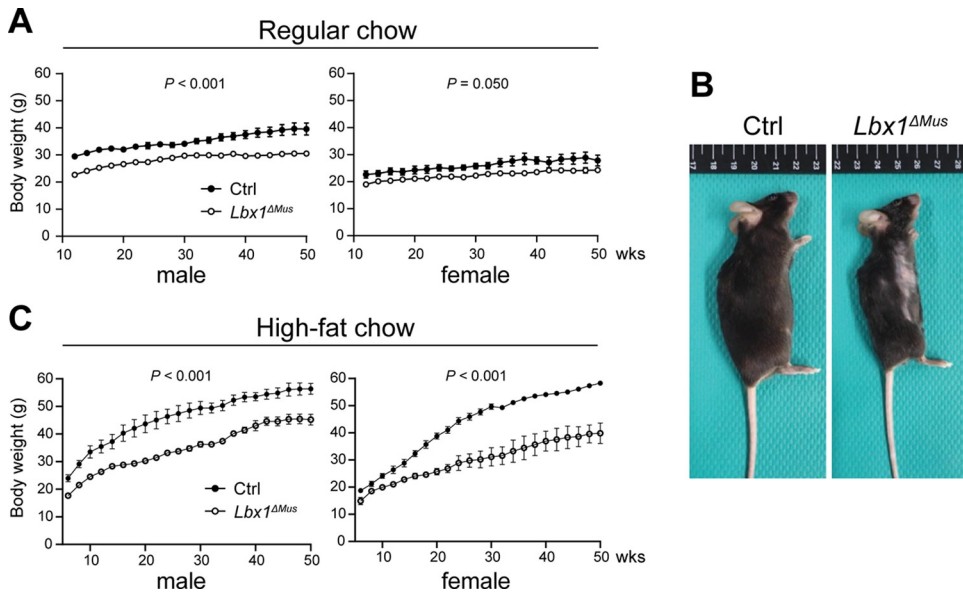

**Fig 1. *Lbx1^ΔMus* mice are resistant to obesity.** (A) Time course of body weight changes in Ctrl and *Lbx1^ΔMus* mice fed regular chow. (B) Representative macroscopic images of 30-week-old female Ctrl and *Lbx1^ΔMus* mice. (C) Time course of body weight changes in Ctrl and *Lbx1^ΔMus* mice fed high-fat chow. $N$ = 3–5 mice per group. Two-way ANOVA, (A) and (C).

## *Lbx1^ΔMus* mice have less fat accumulation than Ctrl mice

Because *Lbx1^ΔMus* mice appeared leaner than Ctrl mice, we hypothesized that *Lbx1^ΔMus* mice accumulate less fat than Ctrl mice. We collected perigonadal adipose tissue from *Lbx1^ΔMus* and Ctrl mice fed regular chow and evaluated fat tissue weight at 11, 30, and 50 weeks of age. Although there was some individual variation, we found that the weight of perigonadal adipose tissue were markedly lower in *Lbx1^ΔMus* mice compared to Ctrl mice (Fig 2A and 2B). On the other hand, there was no significant difference in the weight of the tibialis anterior muscle between *Lbx1^ΔMus* and Ctrl mice (Fig 2C). Histological analysis showed that adipocytes in *Lbx1^ΔMus* mice were markedly smaller and cell membrane appeared more distinct than those in Ctrl mice (Fig 2D). Consistent with this observation, the frequency distribution of adipocyte cross-sectional area showed that adipose tissue of both male and female *Lbx1^ΔMus* mice consisted of smaller adipocytes than that of male and female Ctrl mice (Fig 2E). These observations suggest that *Lbx1^ΔMus* mice are resistant to obesity owing to their propensity to accumulate less fat compared to Ctrl animals.

## Energy metabolism is increased in *Lbx1^ΔMus* mice

Since skeletal muscle is one of the largest glucose-consuming organs in the body, we next performed an intraperitoneal glucose tolerance test using *Lbx1^ΔMus* and Ctrl mice to investigate the potential alteration in glucose metabolism in *Lbx1^ΔMus* mice. We injected the mice with glucose (1.5g / kg) and evaluated the blood glucose level at the designated time points (0, 15, 30, 60, and 120 min). We found that the peak blood glucose level was lower and that blood glucose level decreased to basal levels more rapidly in *Lbx1^ΔMus* mice than in Ctrl mice (Fig 3A). The overall trend was similar between male and female mice; however, the difference reached statistical significance only in females. Evaluation of the area under the curve, which reflects the total increase in blood glucose during a glucose tolerance test, showed a similar result (Fig 3B). In accordance, $VO_2$ max, which reflects the maximum amount of oxygen that can be

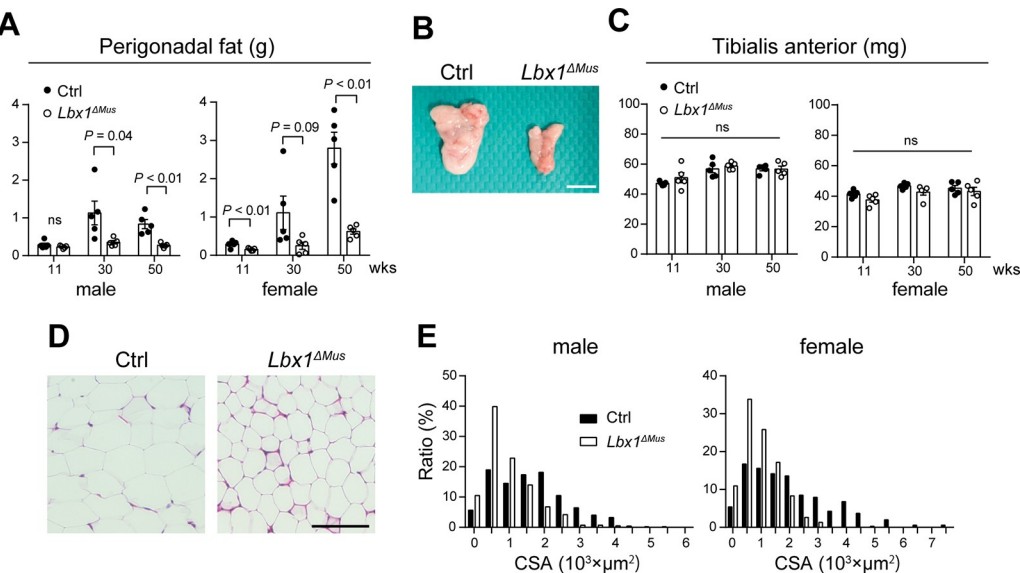

**Fig 2. Smaller adipose tissue in *Lbx1^{ΔMus}* mice compared to Ctrl mice.** (A) Weight of perigonadal fat tissue collected at 11, 30, and 50 weeks of age. *N* = 5 mice per group. (B) Representative macroscopic images of perigonadal fat tissue collected from 50-week-old female Ctrl and *Lbx1^{ΔMus}* mice. Bar, 10 mm. (C) Weight of the tibialis anterior muscle collected at 11, 30, and 50 weeks of age. *N* = 5 mice per group. ns, not significant. (D) Hematoxylin and eosin-stained sections of perigonadal fat tissues of 30-week-old female Ctrl and *Lbx1^{ΔMus}* mice. Bar, 100 μm. (E) Frequency distribution of adipocyte cross-sectional area in perigonadal tissues of 30-week-old Ctrl and *Lbx1^{ΔMus}* mice. Student's *t*-test, (A) and (C).

utilized during exercise, was higher in *Lbx1^{ΔMus}* mice than in Ctrl mice, suggesting that aerobic metabolism is enhanced in *Lbx1^{ΔMus}* mice than in Ctrl mice (Fig 3C). Of note, we found that the core body temperature of *Lbx1^{ΔMus}* mice was significantly higher than that of Ctrl mice under normal conditions (Fig 3D).

Next, basal O2 consumption, respiratory quotient, food intake, and spontaneous activity were assessed using a metabolic cage. While there was no difference in daytime $O_2$ consumption between Ctrl and *Lbx1^{ΔMus}* mice, nighttime $O_2$ consumption tended to be higher in *Lbx1^{ΔMus}* mice than in Ctrl mice in both males and females (Fig 3E; the difference did not reach statistical significance), indicating an increase in energy expenditure in *Lbx1^{ΔMus}* mice. On the other hand, there was no difference in the respiratory quotient between Ctrl and *Lbx1^{ΔMus}* mice (Fig 3F). The leaner body composition in *Lbx1^{ΔMus}* mice could be due to reduced food intake; however, the amount of food consumed per day was comparable between *Lbx1^{ΔMus}* and Ctrl mice (Fig 3G). In fact, there was even a trend toward increased food consumption in male *Lbx1^{ΔMus}* compared to Ctrl mice. Increased activity may also lead to increased energy expenditure and lean body mass; however, on the contrary, both the diurnal and nocturnal activity were significantly lower in *Lbx1^{ΔMus}* mice than in Ctrl mice (Fig 3H). These results indicate that ablation of *Lbx1* in skeletal muscle increases energy expenditure and that neither reduced food intake or increased activity is associated with the leaner body composition in *Lbx1^{ΔMus}* mice.

## LBX1 negatively regulates glucose consumption

To gain insight into the mechanism behind the increased energy metabolism in *Lbx1^{ΔMus}* mice, we performed whole transcript analysis using the tibialis anterior muscle tissues from Ctrl and *Lbx1^{ΔMus}* mice. Among the genes that were differentially expressed between Ctrl and

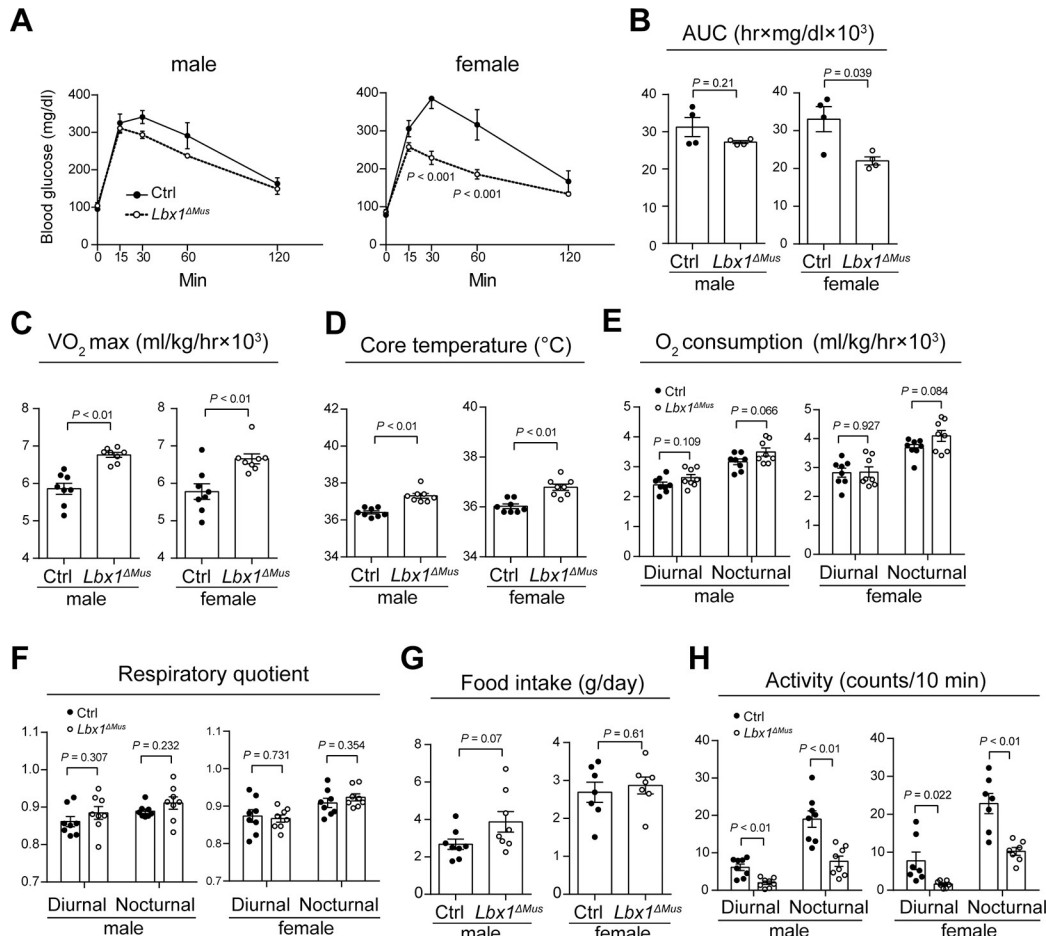

**Fig 3. Increased energy metabolism in *Lbx1^ΔMus* mice.** (A) Time course changes in blood glucose levels after intraperitoneal injection of glucose. (B) The area under the curve (AUC) value of the intraperitoneal glucose tolerance tests presented in Fig 3A. $N$ = 3–5 mice per group. (C-H) VO₂ max (C), core body temperature (D), basal O₂ consumption (E), respiratory quotient (F), food intake per day (G), activity (number of times per 12 h that the subject crossed the center of the cage) (H) of Ctrl and *Lbx1^ΔMus* mice. $N$ = 7–8 mice per group. Two-way ANOVA with Sidak correction, (A); and Student's *t*-test, (B)–(H).

*Lbx1^ΔMus* mice, we found several genes involved in energy metabolism (S1 Fig). We performed quantitative PCR and validated the increased or decreased expression of these genes in the skeletal muscle of *Lbx1^ΔMus* mice and identified *Mss51*, *Igfn1*, *Myh7*, and *Grb14* as candidate genes involved in the increased energy metabolism in *Lbx1^ΔMus* mice (Fig 4A).

We next sought to determine whether the reduced expression of *Lbx1* was associated with increased glucose consumption at the cellular level through these potential target genes. Initially, we attempted to perform gene silencing experiments against *Lbx1*; however, we were unable to identify a mouse cell line that expressed LBX1 protein at detectable levels. Therefore, we performed gain-of-function experiments by introducing an LBX1 expression vector into a non-LBX1 expressing cell line to investigate its potential function in the regulation of glucose metabolism. We generated a Myc-tagged LBX1 expression vector (LBX1^Myc) to facilitate detection and used a mouse fibroblast cell line, NIH/3T3, which does not express LBX1 protein at detectable levels. We confirmed the expression of *Lbx1* transcripts and LBX1^Myc protein as well as its nuclear localization (Fig 4B). As described in the Materials and Methods, cells were transfected with an empty vector or the LBX1^Myc expression vector, and the amount of glucose

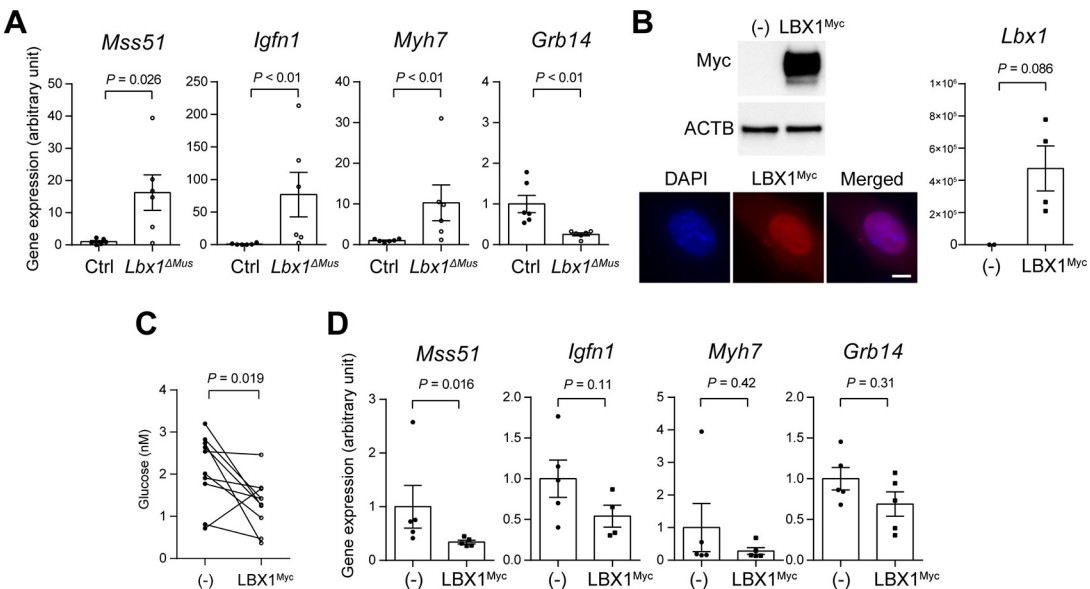

**Fig 4. LBX1 negatively regulates glucose metabolism in a cell-autonomous manner.** (A) Gene expression analysis of the potential downstream targets of LBX1 in skeletal tissue of Ctrl and *Lbx1*<sup>ΔMus</sup> mice. (B) Expression of Myc-tagged LBX1 protein (upper left panel) and transcripts (right panel) in empty vector (-) and LBX1<sup>Myc</sup> expression vector-transfected (LBX1<sup>Myc</sup>) NIH/3T3 cells. Beta-actin (ACTB) is used as an internal control. *N* = 4 independent experiments. Intracellular localization of LBX1<sup>Myc</sup> protein in LBX1<sup>Myc</sup> expression vector-transfected NIH/3T3 cells (lower left panels), Bar 10 μm. (C) Glucose uptake in empty vector (-) and LBX1<sup>Myc</sup> expression vector-transfected NIH/3T3 cells. *N* = 10 independent experiments. (D) Gene expression analysis of the potential downstream targets of LBX1 in empty vector (-) and LBX1<sup>Myc</sup> expression vector-transfected NIH/3T3 cells. Mann-Whitney U test, (A), (B) and (D); and paired t-test, (C).

uptake was compared between them. As shown in Fig 4C, we found that the glucose uptake was significantly reduced in LBX1<sup>Myc</sup>-expressing cells compared to empty vector-transfected cells.

We next investigated whether the introduction of the LBX1<sup>Myc</sup> expression vector could inversely alter the expression of candidate genes potentially involved in the increased glucose consumption in *Lbx1*<sup>ΔMus</sup> mice (Fig 4A). We found that the expression of *Mss51*, *Igfn1*, and *Myh7* transcripts, which was increased in *Lbx1*<sup>ΔMus</sup> mice, was decreased in cells transfected with the LBX1<sup>Myc</sup> expression vector (Fig 4D; note that a statistically significant difference was obtained only for *Mss51*), indicating that LBX1 expression negatively correlates with the transcript levels of these genes. On the other hand, there was no significant increase in the expression of *Grb14* transcripts, which were found to be decreased in *Lbx1*<sup>ΔMus</sup> mice.

Taken together, these observations indicate that overexpression of LBX1 negatively regulates glucose consumption in NIH/3T3 cells potentially through altering the expression of genes involved in energy metabolism, including *Mss51*, *Igfn1*, and *Myh7*.

## Discussion

In this study, we showed that mice lacking LBX1 specifically in skeletal muscle accumulate less fat and are leaner than their age-matched Ctrl counterparts. Of note, we found that these mice are resistant to high-fat diet-induced obesity (particularly in females) despite comparable food intake and lower activity compared to Ctrl mice. In accordance, *Lbx1*<sup>ΔMus</sup> mice showed a lower peak blood glucose level after glucose injection and higher $VO_2$ max and core body temperature compared to Ctrl mice. Furthermore, we found that the expression of several genes involved in energy metabolism and muscle development are potentially dysregulated in

skeletal muscle of $Lbx1^{\Delta Mus}$ mice and that forced expression of LBX1 in NIH/3T3 cells leads to decreased glucose consumption. These results suggest a previously unrecognized role for LBX1 as a negative regulator of energy metabolism in skeletal muscle.

The *LBX1* gene is located close to three SNPs, including rs11190870, associated with susceptibility to AIS and is one of the most promising genes for AIS susceptibility [3, 4]. Among the many functions of LBX1, it has been shown to play an essential role in muscle development and neuronal determination. Several studies have shown that AIS patients have paravertebral muscle asymmetry between the convex and concave sides, indicating that the imbalance in the prevertebral muscles may lead to spinal deformity [12–14]. In accordance, a recent study showed that there are differences in the transcript levels of several genes, including LBX1, between the convex and concave sides of the vertebral muscle in AIS patients [15]. The other line of studies suggests that AIS patients have defects in the proprioceptive system that could potentially impair postural stability and lead to spinal deformity [16, 17]. In support of this idea, studies have shown that LBX1 acts as a fate determinant of somatosensory relay neurons and that mice lacking *Lbx1* have defects in sensory afferent innervation of the dorsal horn [18, 19].

In contrast to these hypotheses suggesting that muscular and/or neuronal defects are associated with the development of AIS, our study may suggest that lean body mass caused by LBX1 dysfunction is another factor associated with susceptibility to AIS. Previous studies have shown that AIS patients are significantly leaner than their age-matched controls [6–10], although the prevalence of eating disorders is lower in AIS patients than in controls [20, 21]. Interestingly, these findings in AIS patients are reminiscent of the phenotype of $Lbx1^{\Delta Mus}$ mice described in the present study. Of note, our data indicate that loss of LBX1 has more pronounced effects in female than in male mice, particularly in glucose sensitivity and resistance to obesity. Given that AIS preferentially affects females over males in humans, this observation may indicate the relevance of this mouse model in studying the pathophysiology of AIS. In addition, previous studies have also suggested that potential dysregulation in lipid, amino acid, and carbohydrate metabolisms is potentially associated with the leanness of AIS patients [22–24]. Although further studies are mandatory to elucidate the causal relationship between lean body mass and the onset of AIS [25, 26], in light of these findings, it is tempting to speculate that defective LBX1 activity increases energy metabolism in humans and that the resulting lean physique leads to increased susceptibility to AIS.

While our data clearly demonstrate that skeletal muscle-specific ablation of LBX1 in mice results in increased energy metabolism and lean body mass, the molecular mechanisms underlying these phenotypes are not fully understood. Gene expression analysis of skeletal muscle from Ctrl and $Lbx1^{\Delta Mus}$ mice identified several candidate genes involved in energy metabolism or muscle development, including *Mss51* [27, 28], *Igfn1* [29], *Myh7* [30], and *Grb14* [31, 32]. Among these, the robust increase in the expression of *Mss51* was particularly intriguing, as it is involved in systemic energy metabolism [27, 28]. We initially attempted to study the function of LBX1 *in vitro* by gene silencing experiments, but this was hampered by the lack of cell lines that express reasonably high levels of LBX1, including the myoblast-like cell line C2C12. Perhaps for the same reason, most studies investigating the function of LBX1 *in vitro* have performed overexpression rather than gene-silencing experiments [24, 33, 34]. Gene expression analysis using LBX1$^{myc}$ vector and empty vector-induced NIH/3T3 cells showed that LBX1 suppressed the expression levels of *Mss51*, *Igfn1*, and *Myh7*; however, the difference was significant only for *Mss51*, potentially due to insufficient transfection efficiency. Nevertheless, we were able to show that overexpression of LBX in NIH/3T3 cells leads to decreased glucose uptake compared to empty vector-induced cells, consistent with the findings in $Lbx1^{\Delta Mus}$ mice that LBX1 negatively regulates glucose metabolism.

This study has several limitations. First, and most critically, the results of the present study need to be interpreted with caution if they could be translated to the pathology of AIS in humans. In our mouse model, LBX1 expression in skeletal muscle is almost completely abolished during embryonic development and postnatal growth [5]. On the other hand, even if LBX1 expression were reduced in humans carrying the risk allele of rs11190870, it would certainly not be reduced to levels comparable to those observed in our mouse model. Furthermore, although the association between rs11190870 and susceptibility to AIS is statistically significant, the odds ratio between the risk allele and non-risk alleles of the SNP is approximately 1.5 to 1.8 [3, 4]. Therefore, even though the risk allele of the SNP could cause a leaner physique, possibly through reduced expression of LBX1, the contribution of the risk allele of this SNP to the onset of AIS is not necessarily robust and could potentially be masked by other genetic and environmental factors at the individual level. Second, total body weight, but not lean body weight, was used to normalize the $VO_2$ max and $O_2$ consumption values (Fig 3C and 3E). We found that the differences in basal O2 consumption between Ctrl and $Lbx1^{\Delta Mus}$ mice were abolished when unadjusted by body weight (S4 Fig). However, considering that $Lbx1^{\Delta Mus}$ mice have underdeveloped forelimbs and therefore less muscle than Ctrl mice [5], that there is no marked difference in fat weight at the time of analysis (11 weeks of age) (Fig 2A), that total activity is significantly lower in $Lbx1^{\Delta Mus}$ mice than in Ctrl mice (Fig 3H), and that there is no difference in food intake (Fig 3G), it is conceivable that total energy expenditure is higher in $Lbx1^{\Delta Mus}$ mice than in Ctrl mice. Last, due to technical difficulties, we performed gain-of-function experiments using the NIH/3T3 fibroblast cell line as an in vitro model, and therefore the results of the cell-based assays in this study may require further validation.

In conclusion, our data show that loss of LBX1 in skeletal muscle leads to resistance to obesity, improved glucose homeostasis, and increased energy metabolism, indicating that LBX1 functions as a negative regulator of energy metabolism. The results of the present study may also suggest that LBX1 dysfunction indirectly increases susceptibility to AIS by altering body mass composition. While this hypothesis requires further investigation, the present study may serve as a basis to explore the potential association of LBX1 and energy metabolism in the etiology of AIS.

## Supporting information

**S1 Fig. Gene expression analysis of skeletal muscle from Ctrl and $Lbx1^{\Delta Mus}$ mice.** Lists of 20 genes found to be most up- (left panel) and down-regulated (right panel) in skeletal muscle tissue of $Lbx1^{\Delta Mus}$ mice compared to Ctrl mice by microarray analysis. Black dots indicate genes with a potential role in energy metabolism whose increased or decreased expression in skeletal muscle was validated by quantitative PCR. The transcriptional expression levels of *Bdh1*, *Igfbp5*, *Irs2*, *Nr4a2*, and *Nr4a3* were also examined, but the difference in expression could not be reproduced by quantitative PCR.
(TIF)

**S2 Fig. Ct values of *Gapdh* transcripts in skeletal muscle harvested form from Ctrl and $Lbx1^{\Delta Mus}$ mice.** $N = 6$. Mann-Whitney U test.
(TIF)

**S3 Fig. Ct values of *Gapdh*, *Mss51*, *Igfn1*, *Myh7*, and *Grb14* transcripts in NIH-3T3 cells and skeletal muscle from Ctrl mice.** $N = 6$.
(TIF)

**S4 Fig. Basal O2 consumption of Ctrl and *Lbx1^{ΔMus}* mice without normalization to body weight.** *N* = 6.
(TIF)

**S1 Raw images. Original gel images used in Fig 4B.**
(PDF)

## Acknowledgments

We would like to thank Takeshi Ono (Department of Global Infectious Diseases and Tropical Medicine, National Defense Medical College), Takemi Oguma (Department of Orthopedic Surgery, National Defense Medical College), Masaki Yoda, and Mika Imamura (Department of Orthopedic Surgery, Keio University School of Medicine) for their technical assistance.

## Author Contributions

**Conceptualization:** Keisuke Horiuchi.

**Data curation:** Keisuke Horiuchi, Yusuke Matsuhashi.

**Formal analysis:** Takahiro Nakagawa.

**Funding acquisition:** Takahiro Nakagawa, Keisuke Horiuchi.

**Investigation:** Takahiro Nakagawa, Kazuki Kagami, Shinya Kondo, Masashi Isaji.

**Methodology:** Takahiro Nakagawa, Keisuke Horiuchi, Yusuke Matsuhashi.

**Resources:** Kazuki Kagami, Takeshi Adachi.

**Supervision:** Keisuke Horiuchi, Kazuya Kitamura, Takeshi Adachi, Kazuhiro Chiba.

**Writing – original draft:** Takahiro Nakagawa, Keisuke Horiuchi.

**Writing – review & editing:** Takahiro Nakagawa, Keisuke Horiuchi, Kazuki Kagami, Shinya Kondo, Masashi Isaji, Yusuke Matsuhashi, Kazuya Kitamura, Takeshi Adachi, Kazuhiro Chiba.

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
