## [Decision Letter · Decision Letter 0]

17 Jun 2024

PONE-D-24-20291Conditional ablation of Lbx1 in skeletal muscle leads to increased energy metabolism and renders resistance to obesity in micePLOS ONE

Dear Dr. Horiuchi,

Thank you for submitting your manuscript to PLOS ONE. After careful consideration, we feel that it has merit but does not fully meet PLOS ONE’s publication criteria as it currently stands. Therefore, we invite you to submit a revised version of the manuscript that addresses the points raised during the review process.

Your manuscript has been reviewed by two experts and they have shown interest in publishing it. However, they have also raised several concerns. Please appropriately address these comments point-by-point.

We look forward to receiving your revised manuscript.

Kind regards,

Keisuke Hitachi

Academic Editor

PLOS ONE

Journal Requirements:

"JSPS KAKENHI (22K09320)."

4. Please expand the acronym “JSPS KAKENHI” (as indicated in your financial disclosure) so that it states the name of your funders in full.

"This work was supported in part by JSPS KAKENHI (22K09320). We would like to thank Takeshi Ono (Department of Global Infectious Diseases and Tropical Medicine, National Defense Medical College), Takemi Oguma (Department of Orthopedic Surgery, National Defense Medical College), Masaki Yoda, and Mika Imamura (Department of Orthopedic Surgery, Keio University School of Medicine) for their technical assistance."

"JSPS KAKENHI (22K09320)."

7. Please amend the manuscript submission data (via Edit Submission) to include author Dr. Yusuke Matsuhashi.

8. We note that you have included the phrase “data not shown” in your manuscript. Unfortunately, this does not meet our data sharing requirements. PLOS does not permit references to inaccessible data. We require that authors provide all relevant data within the paper, Supporting Information files, or in an acceptable, public repository. Please add a citation to support this phrase or upload the data that corresponds with these findings to a stable repository (such as Figshare or Dryad) and provide and URLs, DOIs, or accession numbers that may be used to access these data. Or, if the data are not a core part of the research being presented in your study, we ask that you remove the phrase that refers to these data.

Reviewers' comments:

Reviewer's Responses to Questions

**Comments to the Author**

1. Is the manuscript technically sound, and do the data support the conclusions?

Reviewer #1: Partly

Reviewer #2: Partly

2. Has the statistical analysis been performed appropriately and rigorously? 

Reviewer #1: Yes

Reviewer #2: N/A

3. Have the authors made all data underlying the findings in their manuscript fully available?

Reviewer #1: Yes

Reviewer #2: Yes

4. Is the manuscript presented in an intelligible fashion and written in standard English?

Reviewer #1: Yes

Reviewer #2: Yes

5. Review Comments to the Author

Reviewer #1: The manuscript presents interesting data regarding the function of Lbx1 in muscle metabolism. There are however several concerns which should be addressed to allow clearer conclusions.

Authors hypothesize that increase in energy metabolism resulting from impairment of Lbx1 function can contribute to the development of AIS. It seems more likely that the link would be less straightforward. Considering the role of Lbx1 in muscle development, it is well possible that its impairment results in muscles with impaired energy efficiency consuming more energy than in controls. Such muscles may not be able to secure correct posture leading to AIS. Nevertheless, AIS and energy metabolism would not be causally linked in such a case, they would be rather both caused by the same factor - improper muscle development.

More importantly, to assess the muscle and whole body energy metabolism properly, attention should be paid to the following points:

1. Lbx1 KO mouse are significantly smaller than the controls and the difference is believed to be mostly because of smaller fat mass. If the lean mass is really comparable between both groups, it is misleading to normalize any output to total body weight, as fat mass does not affect much parameters like blood volume or energy expenditure. If both strains are injected with the same dose of glucose, which seems more appropriate in this case, the differences in blood glucose and its AUC would be less pronounced. Similarly, energy expenditure should not be normalized to body weight (or unnormalized data should be present in parallel at least).

2. While VO2max is useful parameter regarding muscle function and strength, basal O2 consumption and 24h O2 consumption would be more relevant to link the data to changes in body weight. Such energy expenditure can be directly compared to energy intake. Moreover, calculation of respiratory exchange ratio would reveal whether the KO mice consume more lipids, further supporting the link to reduced fat mass.

3. The selection of cell line for the cell culture experiments need some justification. For comparison of cell culture experiments to in vivo model of muscle specific Lbx1 manipulation, it would make much more sense to use some muscle cell line.

4. Selection of the genes linked to the energy metabolism is not clear. E.g. Bdh1 (larger fold change than Grb14) is surely involved in metabolism of energy substrates as well.

Minor comments: Authors should make sure that all abbreviations (e.g. AUC, CSA, ACTB) are appropriately explained. Readers would benefit from brief method description of e.g. glucose uptake analysis, cell confluence etc. (besides pointing to manufacturers instructions).

Reviewer #2: The work by Nakagawa and colleagues, "Conditional ablation of Lbx1 in skeletal muscle leads to increased energy metabolism and renders resistance to obesity in mice", could represent a tangible starting point in providing new information useful to improve the understanding of yet unclear aspects of Adolescent Idiopathic Scoliosis (AIS) at both molecular and pathophysiological levels. The authors focus their attention on the transcription factor LBX1, which is highly expressed in spinal cord and skeletal muscle, and whose locus is in proximity of a SNP strongly associated with AIS susceptibility. The authors set up their experimental framework to characterize the role of LBX1 based on previous studies where the ablation of LBX1 in skeletal muscle leads to upper limb hypoplasia and lean, the latter being an aspect often observed in AIS patients as well.

With the aim of verifying whether muscle-specific ablation of LBX1 could alter parameters associated with a leaner body mass, the authors conducted a series of experiments demonstrating that animals with LBX1 selectively ablated in skeletal muscle, exhibit alterations in some of these parameters, mainly related to glucose metabolism. Based also on in vitro experiments performed on NIH-3T3 cells where LBX1 was overexpressed, the authors concluded that the loss of LBX1 is associated with resistance to obesity, improved glucose homeostasis, and increased energy metabolism, and that LBX1 can be considered as a negative regulator of energy metabolism.

Based on the results obtained, it appears evident that LBX1 may play a significant role in regulating energy expenditure in mice and potentially in humans as well. However, there are several aspects related to the manuscript, the interpretation of the experimental data, and some experimental protocols that need to be reviewed and/or better argued/explained before the work can be published:

1) The title does not seem to align with the experimental evidence presented in the work: there is not enough evidence to claim that the ablation of LBX1 "renders resistance to obesity in mice". The data presented to justify this condition are limited to a reduced weight gain in LBX1Δmus mice when subjected to a high-fat diet (HFD). Basic parameters associated with obesity, such as the Lee index, blood lipid and insulin levels, or insulin tolerance in HFD-fed animals, which would more concretely support a condition of "resistance to obesity", were not done. All other parameters analyzed (i.e., perigonadal fat, tibialis muscle weight, food intake, core temperature, and glucose tolerance) pertain to animals fed a standard diet. Additionally, the in vivo experiments on LBX1 ablation and the in vitro experiments on LBX1 overexpression indicate that altering LBX1 expression impacts glucose uptake and the expression of specific genes related to energy metabolism (particularly glucose metabolism). Together with the other in vivo evidence reported, a title more aligned with the obtained results could be: "The alteration of LBX1 expression is associated with changes in parameters related to energy metabolism"

2) In the materials and methods section, it is reported that the HFD treatment begins at 5 weeks. Why, in figure 1, are the weights of the animals shown only starting from 11 or 12 weeks? Additionally, since the duration of the HFD treatment is not indicated in the methods, it is inferred from figure 1C that this treatment lasts for 45 weeks. Why subject the animals to an HFD regime for such a long time? From an animal welfare perspective, the authors should provide adequate experimental justifications, especially considering that no additional experimental procedures other than weighing the animals are performed. In addition, the image presented in figure 1B (11-week-old regular chow-fed females as per the corresponding legend) is not consistent with the data reported in figure 1A (right panel). Such a noticeable difference in body size (1B) is not consistent with an estimated weight difference of 2 or 3 grams (1A). The authors need to better explain this discrepancy.

3) The authors should explain how WT females subjected to an HFD reach a body weight similar to, if not higher than, that of males, when it is known that female mice on an HFD typically achieve a lower absolute weight compared to age-matched males. This significant weight increase in WT females on HFD appears to be anomalous compared to specific literature, potentially suggesting that the genetic modification of the animal colony used in these experiments (Lbx1flox/flox) may not fully represent certain biometric parameters, especially concerning females. A comment from the authors on this issue is expected.

4) Calculating the respiratory quotient (CO2 produced / O2 consumed), in addition to the VO2 value of oxygen consumed, is a metabolic parameter that, based on the study's framework and reported results, seems necessary to better define the type of metabolism in LBX1 Δmus animals (preferentially lipid, preferentially glucose, or balanced between the two).

5) In the materials and methods section, the method for relative quantification of RT-PCR experiments should be specified (2 –ΔΔCT ? ). In a context where authors highlight alterations in energy metabolism, how appropriate is the use of a metabolic enzyme (such as GAPDH) as a normalizer in RT-PCR experiments? Evidence (as supplementary data) should be provided to demonstrate that GAPDH expression remains unchanged in WT and LBX1 Δmus mice.

6) In the discussion, the authors describe Mss1 as a muscle-specific transcription factor. Based on this, how confident can be considered the results of in vitro RT-PCR where LBX1 overexpression reduces Mss1 expression by over 50% compared to control cells (NIH-3T3), where this gene is not expected to be transcribed? To what extent can this consideration be extended to the other genes analyzed in these experiments? The authors should provide data supporting the actual expression of the four genes analyzed in NIH-3T3 cells.

7) A more thorough introduction of LBX1 would help the reader better understand the connections between LBX1, AIS, and the hypothesis that the authors aim to investigate. Specifically, are there data regarding LBX1 expression levels in AIS patients compared to healthy individuals?

6. PLOS authors have the option to publish the peer review history of their article (what does this mean?). If published, this will include your full peer review and any attached files.

Reviewer #1: No

Reviewer #2: No

---

## [Author Response · Author response to Decision Letter 0]

8 Jul 2024

Point-by-point Responses

First, we would like to thank the reviewers for their insightful and constructive critiques of the manuscript. By addressing the issues raised by the reviewers, we believe that our manuscript has substantially improved. 

Reviewer #1: 

1. Lbx1 KO mouse are significantly smaller than the controls and the difference is believed to be mostly because of smaller fat mass. If the lean mass is really comparable between both groups, it is misleading to normalize any output to total body weight, as fat mass does not affect much parameters like blood volume or energy expenditure. If both strains are injected with the same dose of glucose, which seems more appropriate in this case, the differences in blood glucose and its AUC would be less pronounced. Similarly, energy expenditure should not be normalized to body weight (or unnormalized data should be present in parallel at least).

Response: We agree with the reviewer that in some cases it would be preferable to perform GTT experiments using lean body mass rather than total body weight as a reference. However, in the present study, GTT experiments (and other experiments related to energy expenditure) were performed on 11-week-old mice, at which point there is little fat accumulation in Ctrl mice, and the difference in perigonadal fat volume between Ctrl and LbxΔMus mice is negligible (Fig 2A). Therefore, we believe that it would not be completely irrational to use total body weight as a reference in our experiments (the difference in body weight is likely due to the hypoplastic forelimbs, as we have previously described*). Nevertheless, we have included a graph showing basal O2 consumption with and without normalization with body weight (Fig 3E and S4 Fig: please also see the next response). When adjusted for body weight, O2 consumption was slightly higher in LbxΔMus mice than in Ctrl mice, but this difference was abolished when unadjusted. However, considering that total activity is significantly lower in LbxΔMus mice than in Ctrl mice (Fig. 3H) and there is no difference in food intake (Fig 3G), it is conceivable that overall energy expenditure is higher in LbxΔMus mice than in Ctrl mice. 

*Matsuhashi Y, et al. 2022. J Orthop Res 41:884-890

2. While VO2max is useful parameter regarding muscle function and strength, basal O2 consumption and 24h O2 consumption would be more relevant to link the data to changes in body weight. Such energy expenditure can be directly compared to energy intake. Moreover, calculation of respiratory exchange ratio would reveal whether the KO mice consume more lipids, further supporting the link to reduced fat mass.

Response: As requested by the reviewer, we have included figures showing basal O2 consumption and respiratory quotient (Figs 3E and 3F). We found that O2 consumption was comparable or slightly higher (the difference did not reach statistical significance) in LbxΔMus mice compared to Ctrl (especially during the nighttime), even though activity is lower in LbxΔMus mice compared to Ctrl (Fig 3H). On the other hand, we did not observe any difference in the respiratory quotient between Ctrl and LbxΔMus mice.

3. The selection of cell line for the cell culture experiments need some justification. For comparison of cell culture experiments to in vivo model of muscle specific Lbx1 manipulation, it would make much more sense to use some muscle cell line.

Response: We agree with the reviewer that cell culture experiments using NIH3T3 cells are not an ideal substitute for an in vivo model of muscle-specific Lbx1 manipulation. We originally planned to perform gene silencing experiments using the myoblastic cell line C2C12; however, the expression level of Lbx1 transcripts was barely detectable in these cell lines. Furthermore, we also found that even after introducing the LBX1-Myc expression vectors, these cells expressed very low levels of LBX1 protein compared to NIH3T3, possibly due to the low efficiency of lipofection gene transfer. Therefore, we decided that it was not practical to use C2C12 for our experiment. We would also like to point out that C2C12 cells are not a muscle cell line, but a myoblastic-like cell line, and do not express skeletal muscle-related genes in their undifferentiated state. In this respect, even C2C12 (or any myoblastic cell line) may not be a reasonable substitute for an in vivo model of muscle-specific Lbx1 manipulation. We added the following in the Discussion as a limitation of the present study (P. 22); “In addition, due to technical difficulties, we performed gain-of-function experiments using the NIH/3T3 fibroblast cell line as an in vitro model, and therefore the results of the cell-based assays in this study may require further validation”.

4. Selection of the genes linked to the energy metabolism is not clear. E.g. Bdh1 (larger fold change than Grb14) is surely involved in metabolism of energy substrates as well.

Response: The genes selected for analysis in this study (Mss51, Igfn1, Myh7, and Grb14) were those with potential function in energy metabolism and whose increased or decreased expression in skeletal muscle was validated by quantitative PCR. Bdh1 was indeed one of the candidate genes in the first screening, but the difference in its expression level between LbxΔMus and Ctrl did not reach statistical significance, and therefore excluded from further analysis. To avoid misunderstanding, we have added the following in the figure legend for S1 Fig as follows: “Black dots indicate genes with a potential role in energy metabolism whose increased or decreased expression in skeletal muscle was validated by quantitative PCR”. We have also revised the text as follows (p. 16): “We performed quantitative PCR and validated the increased or decreased expression of these genes in the skeletal muscle of Lbx1ΔMus mice and identified Mss51, Igfn1, Myh7, and Grb14 as candidate genes involved in the increased energy metabolism in Lbx1ΔMus mice (Fig 4A)”.

Minor comments: Authors should make sure that all abbreviations (e.g. AUC, CSA, ACTB) are appropriately explained. Readers would benefit from brief method description of e.g. glucose uptake analysis, cell confluence etc. (besides pointing to manufacturers instructions).

Response: To comply with the reviewer’s comments, we have revised the text as follows:

A brief explanation of CSA (P. 7):

The cross-sectional area of adipocytes, which reflects the degree of intracellular fat accumulation, was measured using Image J software (National Institutes of Health, Bethesda, MA).

A brief explanation of AUC (P. 14):

Evaluation of the area under the curve, which reflects the total increase in blood glucose during a glucose tolerance test, showed a similar result (Fig 3B).

A brief explanation of ACTB (P. 33):

Beta-actin (ACTB) is used as an internal control.

Method for analyzing glucose uptake (P. 10):

In brief, cells (1.35 × 105 cells/well) were cultured in 2 ml of maintenance medium (DMEM supplemented with 10%fetal bovine serum and penicillin/streptomycin) in 12-well plates at 37°C. After 16 h of culture, the media were collected and centrifuged at 1,500 rpm for 5 min. The supernatants were incubated with the reaction buffers provided by the manufacturer for 30 min at 37°C, and the absorbance of the supernatant was measured at 450 nm. Glucose consumption was determined by subtracting the pre-culture glucose concentration from the post-culture glucose concentration.

Reviewer #2: 

1) The title does not seem to align with the experimental evidence presented in the work: there is not enough evidence to claim that the ablation of LBX1 "renders resistance to obesity in mice". The data presented to justify this condition are limited to a reduced weight gain in LBX1Δmus mice when subjected to a high-fat diet (HFD). Basic parameters associated with obesity, such as the Lee index, blood lipid and insulin levels, or insulin tolerance in HFD-fed animals, which would more concretely support a condition of "resistance to obesity", were not done. All other parameters analyzed (i.e., perigonadal fat, tibialis muscle weight, food intake, core temperature, and glucose tolerance) pertain to animals fed a standard diet. Additionally, the in vivo experiments on LBX1 ablation and the in vitro experiments on LBX1 overexpression indicate that altering LBX1 expression impacts glucose uptake and the expression of specific genes related to energy metabolism (particularly glucose metabolism). Together with the other in vivo evidence reported, a title more aligned with the obtained results could be: "The alteration of LBX1 expression is associated with changes in parameters related to energy metabolism"

Response: To address the reviewer's comment, the title has been changed to read as follows: "The alteration of LBX1 expression is associated with changes in parameters related to energy metabolism in mice".

2) In the materials and methods section, it is reported that the HFD treatment begins at 5 weeks. Why, in figure 1, are the weights of the animals shown only starting from 11 or 12 weeks? Additionally, since the duration of the HFD treatment is not indicated in the methods, it is inferred from figure 1C that this treatment lasts for 45 weeks. Why subject the animals to an HFD regime for such a long time? From an animal welfare perspective, the authors should provide adequate experimental justifications, especially considering that no additional experimental procedures other than weighing the animals are performed. In addition, the image presented in figure 1B (11-week-old regular chow-fed females as per the corresponding legend) is not consistent with the data reported in figure 1A (right panel). Such a noticeable difference in body size (1B) is not consistent with an estimated weight difference of 2 or 3 grams (1A). The authors need to better explain this discrepancy.

Response: To address the issue raised by the reviewer, we have revised Figure 1C to show the body weight of mice starting at 5 weeks of age. We originally planned to use mice at 45-50 weeks of age, assuming that the difference in energy metabolism between Ctrl and LbxΔMus mice might increase at this age; however, since we did not observe a meaningful difference from the data collected from 45-week-old mice in our preliminary analysis, we decided not to include these data. Regarding animal welfare on feeding high-fat diet, we checked the “GUIDE FOR THE CARE AND USE OF LABORATORY ANIMALS (Eighth Edition)” published by the National Academy of Science; however, we were unable to find any content pertaining to this issue. Most importantly, the protocol had been approved by the Animal Care Committee at our facility and we did not observe any increase in mortality or difficulty in locomotion in the mice fed the high-fat diet during through the course of this project. Taking all of this into account, we could not find any concrete evidence to support the idea that feeding the high-fat diet to mice for 50 weeks was scientifically unethical. Needless to say, we appreciate the reviewer for reminding us of the importance of animal welfare issues when conducting experiments such as ours. With regard to the images shown in Figure 1B, we found that we mistakenly described these mice as 11 weeks old when they were both 30 weeks old. We have corrected this error in the revised manuscript. We apologize for this error and thank the reviewer for pointing it out.

3) The authors should explain how WT females subjected to an HFD reach a body weight similar to, if not higher than, that of males, when it is known that female mice on an HFD typically achieve a lower absolute weight compared to age-matched males. This significant weight increase in WT females on HFD appears to be anomalous compared to specific literature, potentially suggesting that the genetic modification of the animal colony used in these experiments (Lbx1flox/flox) may not fully represent certain biometric parameters, especially concerning females. A comment from the authors on this issue is expected.

Response: There was one wildtype male mouse that was smaller than the others. If we had excluded this mouse from the analysis, the average weight of the male mice would have been higher than the data presented in the study. This is the most likely reason why the average weight of the female mice appeared comparable to that of the male mice. However, since we did not pre-determine the exclusion criteria for the subjects and did not want to arbitrarily manipulate the data, we included all subjects in this experiment. 

4) Calculating the respiratory quotient (CO2 produced / O2 consumed), in addition to the VO2 value of oxygen consumed, is a metabolic parameter that, based on the study's framework and reported results, seems necessary to better define the type of metabolism in LBX1 Δmus animals (preferentially lipid, preferentially glucose, or balanced between the two).

Response: As requested by the reviewer, we have included figures showing O2 consumption and respiratory quotient (Figures 3E and 3F). We found that O2 consumption was comparable or even higher (note that the difference did not reach statistical significance) in LbxΔMus mice compared to Ctrl (especially during the night), even though activity is lower in LbxΔMus mice compared to Ctrl (Figure 3H). On the other hand, we did not observe any difference in the respiratory quotient between Ctrl and LbxΔMus mice.

5) In the materials and methods section, the method for relative quantification of RT-PCR experiments should be specified (2 –ΔΔCT ? ). In a context where authors highlight alterations in energy metabolism, how appropriate is the use of a metabolic enzyme (such as GAPDH) as a normalizer in RT-PCR experiments? Evidence (as supplementary data) should be provided to demonstrate that GAPDH expression remains unchanged in WT and LBX1 Δmus mice.

Response: We agree with the reviewer that choosing the right internal control is always an issue in quantitative RT-PCR experiments. To address the reviewer's concern, we compared the Ct (threshold cycles) values of Gapdh in skeletal muscle between WT and LBX1Δmus mice and found that there was no significant difference and decided that the expression levels of Gapdh transcripts in skeletal muscle between WT and LBX1Δmus mice were comparable (S2 Fig) and could therefore be used as an internal control.

6) In the discussion, the authors describe Mss1 as a muscle-specific transcription factor. Based on this, how confident can be considered the results of in vitro RT-PCR where LBX1 overexpression reduces Mss1 expression by over 50% compared to control cells (NIH-3T3), where this gene is not expected to be transcribed? To what extent can this consideration be extended to the other genes analyzed in these experiments? The authors should provide data supporting the actual expression of the four genes analyzed in NIH-3T3 cells.

Response: We thank the reviewer for bringing this issue to our attention. Because Mss51 was described as a "muscle-specific gene" in a previous study (Fujita R, et al. 2018. FASEB J 32:5012-5025), we took this for granted and did not verify the validity of this statement ourselves. On the Mouse Genome Informatics website (https://www.informatics.jax.org/), we found that transcripts for Mss51 are expressed rather ubiquitously in mice, including the cardiovascular system, connective tissue, nervous system, and respiratory system, with no particular tissue or organ expressing high levels of Mss51. In light of these data, we have decided that it is misleading to describe Mss51 as “a muscle-specific gene” and have removed this phrase from the revised text (P. 19).

Regarding the expression of Mss51 transcripts and the transcripts of the other three genes (Igfn1, Myh7, and Grb14) in NIH-3T3 cells, we checked the Ct value of each gene examined in this study. As shown in S3 Fig, we found that the Ct value of each gene was within a reasonable range (please note that the higher the Ct value of a given gene, the lower its expression level). Although direct comparisons between different genes cannot be made by qPCR, we found that the Ct value for Mss51 was comparable with that of Gapdh in NIH-3T3 cells, indicating that the 

---

## [Decision Letter · Decision Letter 1]

16 Jul 2024

PONE-D-24-20291R1The alteration of LBX1 expression is associated with changes in parameters related to energy metabolism in micePLOS ONE

Dear Dr. Horiuchi,

Thank you for submitting your manuscript to PLOS ONE. After careful consideration, we feel that it has merit but does not fully meet PLOS ONE’s publication criteria as it currently stands. Therefore, we invite you to submit a revised version of the manuscript that addresses the points raised during the review process.

The revised manuscript was evaluated by the original reviewers. One of the reviewers has requested minor revisions. Please respond them adequately before the publication.

We look forward to receiving your revised manuscript.

Kind regards,

Keisuke Hitachi

Academic Editor

PLOS ONE

Journal Requirements:

Reviewers' comments:

Reviewer's Responses to Questions

**Comments to the Author**

1. If the authors have adequately addressed your comments raised in a previous round of review and you feel that this manuscript is now acceptable for publication, you may indicate that here to bypass the “Comments to the Author” section, enter your conflict of interest statement in the “Confidential to Editor” section, and submit your "Accept" recommendation.

Reviewer #1: All comments have been addressed

Reviewer #2: All comments have been addressed

2. Is the manuscript technically sound, and do the data support the conclusions?

Reviewer #1: Yes

Reviewer #2: Yes

3. Has the statistical analysis been performed appropriately and rigorously? 

Reviewer #1: Yes

Reviewer #2: I Don't Know

4. Have the authors made all data underlying the findings in their manuscript fully available?

Reviewer #1: Yes

Reviewer #2: (No Response)

5. Is the manuscript presented in an intelligible fashion and written in standard English?

Reviewer #1: Yes

Reviewer #2: Yes

6. Review Comments to the Author

Reviewer #1: Authors addressed all the comments and included additional data (mostly as Supplementary figures). However, the supplementary figures are not commented in the main text. S4 Figure is not even included in the List of Supporting information.

The new data (mean energy expenditure, respiratory quotient) rather do not support the original hypothesis that limited weight gain is due to increased energy expenditure. These parameters do not show significant differences and the tendencies can be completely reverted by different way of normalization (per mouse vs per body weight) as discussed in Author´s response but not in the new version of manuscript. If these limitations are properly disclosed, the reader can make more balanced conclusion based on the presented data.

Authors also clarify that genes in focus were selected based on "a potential role in energy metabolism" and differences in expression "validated by quantitative PCR". It would imply that several more (or even all of?) genes from Figure S1 were checked by qPCR, but only few of them were successfully validated. If this is the case, than the list of primers should include primers for all the genes tested and not only those successfully validated.

Reviewer #2: The authors have satisfactorily responded to all my questions and made the necessary changes to the manuscript.

7. PLOS authors have the option to publish the peer review history of their article (what does this mean?). If published, this will include your full peer review and any attached files.

Reviewer #1: No

Reviewer #2: No

---

## [Author Response · Author response to Decision Letter 1]

18 Jul 2024

Reviewer #1: 

1. Authors addressed all the comments and included additional data (mostly as Supplementary figures). However, the supplementary figures are not commented in the main text. S4 Figure is not even included in the List of Supporting information.

Response: We apologize for inadvertently omitting S4 Fig from the list of supporting information. We have added it to the revised manuscript. Regarding the citation of supplementary materials in the text, the PLOS ONE instructions indicate that they do not need to be cited in the text (see below).

“We recommend that you cite supporting information in the manuscript text, but this is not a requirement. Cite the files using the format outlined in Item Description. If you cite supporting information in the text, citations do not need to be in numerical order.”

https://journals.plos.org/plosone/s/supporting-information#loc-item-description

2. The new data (mean energy expenditure, respiratory quotient) rather do not support the original hypothesis that limited weight gain is due to increased energy expenditure. These parameters do not show significant differences and the tendencies can be completely reverted by different way of normalization (per mouse vs per body weight) as discussed in Author´s response but not in the new version of manuscript. If these limitations are properly disclosed, the reader can make more balanced conclusion based on the presented data.

Response: To address the reviewer’s concern, we have included the following as a limitation in the Discussion section (p. 22): “Second, total body weight, but not lean body weight, was used to normalize the VO2 max and O2 consumption values (Figs 3C and 3E). We found that the differences in basal O2 consumption between Ctrl and Lbx1ΔMus mice were abolished when unadjusted by body weight (S4 Fig). However, considering that Lbx1ΔMus mice have underdeveloped forelimbs and therefore less muscle than Ctrl mice, that there is no marked difference in fat weight at the time of analysis (11 weeks of age) (Fig 2A), that total activity is significantly lower in Lbx1ΔMus mice than in Ctrl mice (Fig. 3H), and that there is no difference in food intake (Fig 3G), it is conceivable that total energy expenditure is higher in Lbx1ΔMus mice than in Ctrl mice.”

Regarding the respiratory quotient, our understanding is that it does not reflect the level of total energy expenditure, but only the ratio of energy pathways used for fats, carbohydrates, and proteins. Therefore, even if there were no differences in respiratory quotient values between Ctrl and Lbx1ΔMus mice, we cannot conclude that there was no difference in energy expenditure. 

Authors also clarify that genes in focus were selected based on "a potential role in energy metabolism" and differences in expression "validated by quantitative PCR". It would imply that several more (or even all of?) genes from Figure S1 were checked by qPCR, but only few of them were successfully validated. If this is the case, than the list of primers should include primers for all the genes tested and not only those successfully validated.

Response: We have included the list of genes we examined in Table 1. We have also added a brief comment in the Figure legend (S1 Fig) as follows: “The transcriptional expression levels of Bdh1, Igfbp5, Irs2, Nr4a2, and Nr4a3 were also examined, but the difference in expression could not be reproduced by quantitative PCR.”

Reviewer #2: The authors have satisfactorily responded to all my questions and made the necessary changes to the manuscript.

Response: We are pleased to learn that the concerns raised by the reviewer have been properly addressed. We would like to thank the reviewer again for her/his constructive comments.

---

## [Decision Letter · Decision Letter 2]

24 Jul 2024

The alteration of LBX1 expression is associated with changes in parameters related to energy metabolism in mice

PONE-D-24-20291R2

Dear Dr. Horiuchi,

We’re pleased to inform you that your manuscript has been judged scientifically suitable for publication and will be formally accepted for publication once it meets all outstanding technical requirements.

Kind regards,

Keisuke Hitachi

Academic Editor

PLOS ONE

Additional Editor Comments (optional):

Reviewers' comments:

Reviewer's Responses to Questions

**Comments to the Author**

1. If the authors have adequately addressed your comments raised in a previous round of review and you feel that this manuscript is now acceptable for publication, you may indicate that here to bypass the “Comments to the Author” section, enter your conflict of interest statement in the “Confidential to Editor” section, and submit your "Accept" recommendation.

Reviewer #1: All comments have been addressed

2. Is the manuscript technically sound, and do the data support the conclusions?

Reviewer #1: Yes

3. Has the statistical analysis been performed appropriately and rigorously? 

Reviewer #1: Yes

4. Have the authors made all data underlying the findings in their manuscript fully available?

Reviewer #1: Yes

5. Is the manuscript presented in an intelligible fashion and written in standard English?

Reviewer #1: Yes

6. Review Comments to the Author

Reviewer #1: I thank authors for including the additional information. My suggestions were addressed sufficiently.

7. PLOS authors have the option to publish the peer review history of their article (what does this mean?). If published, this will include your full peer review and any attached files.

Reviewer #1: No

---

## [Editor Report · Acceptance letter]

29 Jul 2024

PONE-D-24-20291R2 

PLOS ONE

Dear Dr. Horiuchi, 

I'm pleased to inform you that your manuscript has been deemed suitable for publication in PLOS ONE. Congratulations! Your manuscript is now being handed over to our production team.

Kind regards, 

on behalf of

Dr. Keisuke Hitachi 

Academic Editor

PLOS ONE